# Protein Enrichment of Wheat Bread with Microalgae: *Microchloropsis gaditana*, *Tetraselmis chui* and *Chlorella vulgaris*

**DOI:** 10.3390/foods10123078

**Published:** 2021-12-10

**Authors:** Waqas Muhammad Qazi, Simon Ballance, Katerina Kousoulaki, Anne Kjersti Uhlen, Dorinde M. M. Kleinegris, Kari Skjånes, Anne Rieder

**Affiliations:** 1Nofima AS—Norwegian Institute of Food, Fisheries and Aquaculture Research, PB 210, NO-1431 Ås, Norway; waqqasqazy@gmail.com (W.M.Q.); simon.ballance@nofima.no (S.B.); anne.uhlen@nmbu.no (A.K.U.); 2Nofima AS—Norwegian Institute of Food, Fisheries and Aquaculture Research, PB 1425 Oasen, NO-5844 Bergen, Norway; katerina.kousoulaki@nofima.no; 3Department of Plant Sciences, Norwegian University of Life Sciences, PB 5003, NO-1432 Ås, Norway; 4NORCE Norwegian Research Centre, Thormøhlensgate 53, NO-5006 Bergen, Norway; dokl@norceresearch.no; 5Department of Biological Sciences, University of Bergen, Thormøhlensgate 53, NO-5006 Bergen, Norway; 6Division of Biotechnology and Plant Health, Norwegian Institute of Bioeconomy Research (NIBIO), PB 115, NO-1431 Ås, Norway; kari.skjanes@nibio.no

**Keywords:** dough rheology, bread-quality, protein-quality, microalgae, *Microchloropsis gaditana*, *Tetraselmis chui*, *Chlorella vulgaris*

## Abstract

Cell wall disrupted and dried *Microchloropsis gaditana* (Mg), *Tetraselmis chui* (Tc) and *Chlorella vulgaris* (Cv) microalgae biomasses, with or without ethanol pre-treatment, were added to wheat bread at a wheat flour substitution level of 12%, to enrich bread protein by 30%. Baking performance, protein quality and basic sensory properties were assessed. Compared to wheat, Mg, Tc and Cv contain higher amounts of essential amino acids and their incorporation markedly improved protein quality in the bread (DIAAS 57–66 vs. 46%). The incorporation of microalgae reduced dough strength and bread volume and increased crumb firmness. This was most pronounced for Cv and Tc but could be improved by ethanol treatment. Mg gave adequate dough strength, bread volume and crumb structure without ethanol treatment. To obtain bread of acceptable smell, appearance, and colour, ethanol treatment was necessary also for Mg as it markedly reduced the unpleasant smell and intense colour of all algae breads. Ethanol treatment reduced the relative content of lysine, but no other essential amino acids. However, it also had a negative impact on *in vitro* protein digestibility. Our results show that Mg had the largest potential for protein fortification of bread, but further work is needed to optimize pre-processing and assess consumer acceptance.

## 1. Introduction

Certain green microalgae (Chlorophyta) can play a role as potential responsible sources of high-quality protein for human nutrition [1]. One possible method is to incorporate these proteins into staple foods, such as bread. Bread has a high content of starch, i.e., ~42–56% [2,3] but contains only low levels of protein with low levels of some essential amino acids, such as lysine. Bread would, therefore, benefit from fortification with microalgae protein. Selected species of microalgae can not only reach high protein contents (40–65% in dry matter) but offer also high-quality proteins [4]. The essential amino acid index (EAAI) gives the geometric mean of contents of each essential amino acid relative to egg protein [5] and is a common measure of protein quality. High EAAI of 0.89 to 1.02 have been reported for *Nannochloropsis gaditana* (recently re-named to *Microchloropsis gaditana*), *Chlorella vulgaris* and *Tetraselmis chui*. Slightly lower EAAI (0.81) has been reported for the edible cyanobacteria *Spirulina platensis*, (sometimes also referred to as a microalga) [6].

The use of microalgae and cyanobacteria to improve the nutritional status of bread has been explored in different studies; with *Spirulina platensis* [2,3,7], *Chlorella vulgaris* [8], *Tetraselmis chui* [9,10] and *Nannochloropsis gaditana* [11,12]. Common for most studies, on the enrichment of bread with microalgae or cyanobacteria, is the small addition or wheat flour replacement level (1–5%), which are not sufficient for a big improvement in protein content or quality. To fulfill the EFSA (European Food Safety Authority, Pama, Italy) specific guidelines for the nutrition claim “increased protein content” a product must provide a minimum of 12% of its energy from proteins (“source of protein”) and contain 30% more proteins compared to a similar product (non-enriched bread). In a recent study using cell wall disrupted *Tetraselmis chui* with a protein content of 50–60%, wheat flour substitution levels of 12% increased the protein content of bread sufficiently for an “increased protein content” nutrition claim [13].

While low doses of addition (≤3%) only slightly impact bread quality [8,9,11], higher microalgae doses (5–10%) lead to inferior bread quality (lower volume, denser crumb, dark green colour, negative effects on taste and smell) and such effects can be quite substantial [3,13,14]. The incorporation of microalgae influences dough rheology, which results in low bread quality (reduced bread volume and increased crumb firmness) [13]. Increasing levels of wheat flour substitution can also result in sticky doughs that create difficulties in machinability due to reduced elasticity [15]. Since each algae species has a unique macronutrient composition, they are expected to influence dough structure, nutritional quality and sensory attributes of bread in different ways. It is, therefore, important to select the most suitable algae species of high nutritional value, and without detrimental effects on dough rheology, for fortification of wheat bread.

Only some microalgae and cyanobacteria species are defined as foods by EFSA and can thus be incorporated and consumed *ad libitum*. This includes the cyanobacterium *Spirulina platensis* and the green microalga *Chlorella vulgaris*. Other species, such as the diatom *Odontella aurita* and the green microalga *Tetraselmis chui* have been approved as novel foods by EFSA but are restricted to specific applications and daily intake levels [16]. For *Nannochloropsis gaditana* (now *Microchloropsis gaditana*) a novel food approval application has been submitted [17].

Microalgae acquire their dark green colour from pigments (chlorophyll). Some studies [7,13,18] have eliminated these pigments by extraction with ethanol. In a baguette, prepared with 1% ethanol-treated *Spirulina platensis*, sensory properties (colour, taste, and smell) were improved compared to the baguette with untreated algae [7]. Another study [18] claimed that incorporation of ethanol-treated *Spirulina platensis* and *Oscillatoria amphibia* (addition level 6.9%) improved dough rheology and baking performance compared to bread without microalgae. Ethanol treatment of *Tetraselmis chui* removed pigments, fats, and flavour compounds, subsequently increasing the protein content [13]. This minimized the negative impact of *Tetraselmis chui* incorporation on dough structure.

The aim of the present study was to compare the application potential of three different microalgae species with high protein content and quality for the protein enrichment of wheat bread by replacing 12% of the wheat flour. The three species *Microchloropsis gaditana* (Mg), *Chlorella vulgaris* (Cv) and *Tetraselmis chui* (Tc) were grown for optimal protein yield, subjected to cell wall disruption, and used either dried or after extraction with ethanol. The impact of the microalgae on dough rheology, bread characteristics and protein quality of the fortified bread was assessed.

## 2. Materials and Methods

### 2.1. Materials

Wheat flour with high protein strength was purchased from Lantmännen (Lantmännen Cerealia, Oslo, Norway). According to the manufacturer, the wheat flour contained 11 g protein, 2 g fat, 3.3 g dietary fibre, 2 g ash and 14.5 g moisture per 100 g flour.

#### 2.1.1. Cultivation of Microalgae

The freshwater green microalga *Chlorella vulgaris* SAG 211-11b (Cv) was obtained from the SAG Culture Collection of Algae (Göttingen, Germany). The stock cultures were maintained using a modified TAP medium [19] where acetate was omitted, at 22 °C and at a light intensity of ~80 µmol m^−2^ s^−1^. Upscaling of the culture was performed using M8a nutrient medium [20]. The biomass was produced in 250 L horizontal tube photobioreactors from LGem BV (de Kwakel, The Netherlands) with LED lights (EAX200 5000 K from Evolys AS, Oslo, Norway), placed indoors in a temperature regulated room, using semi-continuous cultivation and harvesting 90% of the culture every 4–6 days. The cultivation was performed with temperature fluctuation between 25–27 °C, and stepwise increasing light intensity from 600–800–1200 µmol m^−2^ s^−1^ according to culture density. The culture density at the time of harvesting was 3.0 ± 1.3 g DW l^−1^. The algae biomass was harvested by centrifugation at 3000× *g* using an Evodos 10 centrifuge (Evodos BV, Raamsdonksveer, The Netherlands). The resulting algae paste had ~15–20% dry weight and was stored at −20 °C until further processing.

The marine microalga species *Microchloropsis gaditana* CCMP526 was obtained from Bigelow NCMA, and the marine microalga species *Tetraselmis chui* UTEX LB232 was obtained from the UTEX culture collection. For cultivation modified WUR medium was used, based on sterilized natural seawater (salinity 35 ppt) enriched with a nutrient stock solution resulting in the following concentrations (in mM): NaNO3, 25; KH_2_PO4, 1.7; Na_2_EDTA, 0.56; Fe_2_SO_4_·7H_2_O, 0.11; MnCl_2_·2H_2_O, 0.01; ZnSO_4_·7H_2_O, 2.3 × 10^−3^; Co(NO_3_)_2_·6H_2_O, 0.24 × 10^−3^; CuSO_4_·5H_2_O, 0.1 × 10^−3^; Na_2_MoO_4_·2H_2_O, 1.1 × 10^−3^. Both species were cultivated in 250 mL Erlenmeyer flasks at 15 °C and a 16:8 h light-dark cycle (50 µmol m^−2^ s^−1^). This step was followed by cultivation in 25 L tubular photobioreactor (LGEM, de Kwakel, The Netherlands). These cultures were used to inoculate a 250 L photobioreactor (LGEM, de Kwakel, The Netherlands) located indoors in a climate room, where the temperature was kept at 23.0 ± 0.5 °C. The pH was kept at 7.8, through controlled pulse-wise sparging of 100% CO_2_ in the ingoing airstream. Artificial illumination was provided through fluorescent light tubes (PHILIPS Master, TL-D, HF, 50W/840) located inside the tubular helix set-up. The incident light intensity at the start of the cultivation was 30 µmol m^−2^ s^−1^ and was increased stepwise up to 300 µmol m^−2^ s^−1^. The reactor was operated in dual mode, as such mixing was provided by both liquid pump and air pump, resulting in a liquid velocity of approximately 0.3 m s^−1^. This system was run in fed-batch mode, and culture was harvested and transported to Mongstad for inoculation of the large-scale photobioreactors at the National Algae pilot Mongstad, in Norway (60.803230, 5.026794). Here, biomass was produced in four 800 L tubular photobioreactors (LGEM, de Kwakel, The Netherlands), located in a greenhouse and as such exposed to natural light and are additionally equipped with artificial illumination (EAX 170 W LED lights, Evolys AS, Oslo, Norway) with an average incident artificial light of 220 µmol m^−2^ s^−1^. The reactors were operated at pH 7.8 by on-demand CO_2_ addition, and culture temperatures were maintained between 15 and 35 °C by heating the greenhouse or spraying the reactors with water, respectively. The reactors were operated in dual mode, as such mixing was provided by both liquid pump and air pump, resulting in a liquid velocity of approximately 0.3 m s^−1^. The algae were produced in a fed-batch process: the reactors were harvested once per week (between 50–90% of the culture volume), after which seawater and nutrients were added to compensate for the volume taken. The biomass has been produced in various batches in the periods August–October 2017 and April–May 2019. After harvesting, the biomass was dewatered using a spiral plate centrifuge (Evodos 25, Evodos BV, Raamsdonksveer, The Netherlands), resulting in a paste of approximately 25% dry weight for Ng, and 35% dry weight for Tc. The paste was vacuum packed and directly frozen at −23 °C before further processing.

#### 2.1.2. Taxonomic Identification

The taxonomy of the grown microalgae species was confirmed by molecular nuclear marker analysis of the 18S rRNA gene. For *Tetraselmis* sp. a 100% match in the Basic Local Alignment Search Tool (BLAST) was obtained for 18S rDNA oligonucleotide primers ss5 (F) and ss3 (R) and primers 18SF Forward and 18SR Reverse [21] with 651 base pairs (bp) sequenced for each. For the *Microchlroropsis gaditana* sample, the search was conducted against *Nannochloropsis* sp. and a 99.84% and 99.49% BLAST match was respectively obtained for 18SF + 18SF with 621 bp sequenced, and Ss5 with 778 bp sequenced. For *Chlorella* sp. a 100% and 99.74% BLAST match was respectively obtained with 631 bp sequenced for 18SF + 18SR and 772 bp sequenced for Ss3.

#### 2.1.3. Cell Wall Disruption of Microalgae by Bead Milling

To increase the nutrient availability of the microalgae, all three species were subjected to bead milling for cell wall disruption. The bead milling conditions were chosen based on optimization trials for Tc, Mg [22] and Cv and bead milling was performed in a Dyno-Mill Multi Lab (Willy A. Bachofen, Muttenz, Switzerland) at an 80% chamber filling rate in a 1.4 L milling chamber operated at 12 m/sec tip speed (2865 rpm) and a processing temperature of ca. 26 °C.

For Cv, approximately 92% cell wall disruption was achieved at a 13.2 kg/h flow rate using biomass of approximately 18.5% dry matter and zirconium beads of 0.3 mm diameter. Tc paste was diluted to ca. 21% dry matter for bead milling. Bead milling of Tc was performed using 0.25–0.4 mm diameter glass beads by a single pass in a 1.4 L bead mill chamber at 2865 rpm (12 m/s) tip speed and 9.4 kg/h biomass flow rate, and processing temperature of ca. 26 °C, reaching 97% cell wall disruption degree. Cell wall disruption efficiency in Mg was found to be significantly affected only by flow rate, and not by tip speed or biomass dry matter, reaching ~86% disruption at the lowest flow rate tested. On the other hand, soluble protein release increased by increasing biomass dry matter at lower flow rates [21]. Thus, for the disruption of the test Mg biomass used in this study, we used optimized bead milling settings based on a previous trial performed with *Microchloropsis* sp. CCAP 211/78 biomass using a biomass with 18% dry matter, in a 1.4 L milling chamber filled with 0.3 mm diameter Zr beads and running at 2865 rpm tip speed and approx. 9 kg/h flow rate and obtained biomass of approx. 84% cell wall disruption degree. Cell wall disruption degree was estimated in freshly thawed aliquots. by cytometry in a Neubauer counting chamber, observed in a Nikon eclipse Ci optical microscope.

#### 2.1.4. Drying of Cell-Wall Disrupted Microalgae Biomass

Drying was performed at the Aquaculture Feed Technology centre of Nofima in Bergen, Norway. Cell wall disrupted biomass of Cv was spray-dried by a Standard P-6,3 rotary atomizer (GEA Niro AS, DK; Niro Atomizer, Gladsaxe, Denmark) with inlet temperature 220 to 225 °C and outlet temperature 94 to 100 °C. Mg and Tc were dried by a Christ Gamma 1–16 LSC freeze drier at 0.37 mbar. The freeze dryers ice condenser coil temperature was approx. −48 °C and the shelf temperature was set at 25 °C. The biomass ice temperature during freeze-drying was approx. −30 °C. The cell wall disrupted and dried biomass of the three different algae species were termed TcR, CvR and MgR.

### 2.2. Soxhlet Treatment with Ethanol

The raw biomasses of the three microalgae species (TcR, CvR, MgR) were treated with 96% ethanol using a Soxhlet extractor apparatus (Adams & Chittenden Scientific Glass, Berkeley, CA, USA). TcR and CvR, which consisted of small particles (<1 mm), were placed in a cellulose thimble (~50–70 g), MgR, which consisted of larger particles (>1 mm), was placed directly in a glass thimble with a glass sinter filter in the base (80–100 g). The thimbles were placed inside the extraction chamber. To condense the evaporated ethanol, a condenser was attached to the top of the extraction chamber. Water held at 9 °C was circulated through the condenser. A volume of 500 mL ethanol was placed in the boiling flask and heated with a stirred silicon oil bath set at 150 °C. Each batch was continuously extracted until the ethanol in the extraction chamber was colourless, which took approximately 65 h for Tc, 54 h for Cv and 43 h for Mg. The extracted biomasses were spread on a large tray and placed at 75 °C for 6 h in a ventilated oven (Termaks, Bergen, Norway) to remove traces of remaining ethanol and were termed TcT, CvT and MgT.

### 2.3. Compositional Analysis

Moisture content was determined by drying the samples at 105 °C to a constant weight (ICC 109/01). The ash content was determined gravimetrically as residue after combustion in a muffle furnace to 550 °C following AACC 08-01. Amino acid composition of wheat flour, microalgae and bread was estimated after hydrolysis in 6 M HCl for 22 h at 110 °C and analysed using HPLC and fluorescence detection [23]. The protein content of wheat flour, microalgae and bread are reported as the sum of all protein-bound amino acids. For protein digestibility determination [24] the protein content of the bread and digests was determined by combustion [25] using a Vario EL elemental analyser (Elementar, Langenselbold, Germany) with a nitrogen to protein conversion factor of 5.7 [26]. Free amino acids were determined by homogenizing the sample in an internal standard solution, derivatization with phenyl isothiocyanate, separation with reverse HPLC and detection by UV [27]. Crude fat content was estimated by a gravimetric method using solvent extraction [28]. The total dietary fibre (TDF) was measured gravimetrically following AOAC method 985.29 using an Ankom dietary fibre analyser. At least duplicate analyses were performed for all measurements.

### 2.4. Assessment of Protein Quality

The amino acid composition of the algae biomasses before and after ethanol treatment was evaluated by the essential amino acid index (EAAI) calculated according to Oser [5]. The EAAI gives the geometric mean of the ratios of each EAA in a protein to that in a reference protein (egg). High-quality proteins have an EAAI > 0.95 (1 for the reference), while EAAI scores between 0.86 to 0.95 are still considered good [6]. Digestible indispensable amino acid score (DIAAS) values for ingredients (flour and microalgae), as well as fortified bread, were calculated as described by FAO using the reference pattern values for adults [29]. True ileal digestibility values for individual amino acids in microalgae and fortified bread are not available and were, therefore, estimated *via* published data. For wheat flour and bread, data from pigs with ileorectal anastomosis fed a wheat-based diet was used [30]. For microalgae, data for *Spirulina platensis* determined in humans by dual stable isotope tracing were used [31].

### 2.5. In Vitro Protein Digestibility

Protein digestibility of raw and ethanol-treated algae and fortified bread was estimated using a standardized, static, international consensus *in vitro* digestion model (INFOGEST) [32] combined with measurements of protein solubility (combustion) and peptide size distributions generated during simulated digestion [24]. Digestions were performed in triplicates as previously described [24] and sample amounts per tube were standardized according to protein content (170 mg protein per tube). Peptide size distributions in supernatants after simulated digestion were analysed using size exclusion chromatography (SEC) with UV detection as previously described [24,33]. Combined with measurements of protein concentrations in supernatants by combustion, the information in the SEC chromatograms was used to express protein digestibility as the proportion of small soluble peptides (1 kDa and smaller) generated during simulated digestion [24].

### 2.6. Dough Rheology

The mixing properties of wheat flour substituted with 12% (*w*/*w*) algae biomass (TcR, CvR, MgR, TcT, CvT or MgT) were determined using a Farinograph NewPort Scientifc DoughLab (Perten Instruments, Stockholm, Sweden) equipped with a 50 g bowl. Water absorption (WA) and dough stability time (DST) was measured following ISO 5530-1 (1997) method (63 rpm, 30 °C, 500 FU) based on a flour moisture content of 14% (*n* = 2 or more). For dough rheology measurements, doughs were prepared with 1.5% salt (based on flour weight) and otherwise prepared and analysed with Extensograph (resting time 90 min) and Rheometer (resting time 30 min) as previously described [13]. For creep recovery, the sample was subjected to creep (constant stress σ = 100 Pa) applied for 500 s followed by a sudden release of the stress (σ = 0 Pa). Recovery of the sample was measured for 1000 s.

### 2.7. Baking

Wheat flour was replaced by 12% *w*/*w* algal biomasses TcR, CvR, MgR, TcT, CvT, or MgT. Doughs were prepared in duplicates and in random order and baked as previously described [13]. Bread weight, volume, crumb characteristics and crumb firmness were measured as previously described [13]. Bread prepared with 12% TcR and CvR were coated with white icing made of icing sugar and water before volume measurement to avoid light scattering of the reflected laser light on the glossy surface of these breads. Weights were recorded without icing.

### 2.8. Colour Measurement

The colour of algal biomass and bread crumb was measured with a Minolta colourimeter (CR 400, Tokyo Japan) following CIELAB system: L*—lightness (0 black to 100 white); a* green to red (−60 to 60) and b* blue to yellow (−60 to 60). The results presented are based on three replicates on each slice of the three breads for two different baking sessions, i.e., *n* = 18. The total colour difference (∆*E*) was estimated from Equation (1):(1)$$\Delta E=\sqrt{{\left(\Delta L^{*}\right)}^{2}+{\left(\Delta a^{*}\right)}^{2}+{\left(\Delta b^{*}\right)}^{2}}$$

### 2.9. Preliminary Sensory Assessment

Microalgae fortified bread was evaluated by six untrained panellists aged 37–65. Each bread sample was presented to panellists in random order. Water was served between the samples. The bread was ranked for sensory attributes based on colour, smell and overall appearance using a questionnaire based on a nine-point hedonic scale (from −4 to 4). The panellists were requested to write supplementary remarks related to the tested products. The ranking of each bread is presented as the average score by all panellists.

### 2.10. Statistical Analysis

Statistical analysis was performed using Analysis of Variance (ANOVA). Prior to ANOVA, the data were tested for normal distribution (which was true for all the data) and equal variance using Levene’s test. Significant differences between means were analysed using the Tukey posthoc test (alpha = 0.05) in Minitab version 19. Pearson correlation between rheological measurements and the bread crumb properties was performed in Microsoft Excel, version 2016. All rheological measurements and measurements of bread properties were based on at least duplicate doughs. Baking was performed with duplicate doughs and in random order.

## 3. Results and Discussion

### 3.1. Nutrient Composition of Microalgae Samples

All three algae samples had a high protein content (>42%) (Table 1), which increased further after ethanol treatment (>58%). Protein content for Mg (43.3%) was higher than previously reported (30.29%) [6], while protein contents for Tc and Cv were similar to previously published values [1,4]. Tc had a very high mineral (ash) content (16.7%), which is typical for this species [4]. The main effect of ethanol treatment was an almost complete removal of fat and fat-soluble components with a subsequent proportional increase in protein. Ethanol treatment had little effect on mineral content (ash) but reduced the amount of free amino acids in the samples, especially for Cv and Mg.

The microalgae were very different in colour attributes. CvR powder had the darkest colour in this study (L* = 12.9) in comparison to TcR and MgR. Ethanol treatment of TcR, CvR and MgR resulted in a significantly lighter colour, depicted in higher L* values in the corresponding treated algae TcT, CvT and MgT. Ethanol treatment also resulted in a significant reduction in green colour in all three algae. Complete elimination (positive a* values) of the green colour was noticed in CvT and MgT, while treatment with ethanol did not remove all the green pigments in Tc, a* = −10.1 vs. a* = −5.6 in TcR and TcT, respectively. The blue colour was absent in all the samples, and they were dominated by the yellow hue (positive b* values), which increased significantly with ethanol treatment. An overall colour difference, ∆*E* > 70 was noticed with Tc, Cv and Mg compared to wheat flour. With ethanol treatment a reduction in ∆*E* value over 40% was achieved in the corresponding TcT, CvT and in the MgT.

### 3.2. Amino Acid Composition and Assessment of Protein Quality in the Microalgae

Compared to wheat flour, all algae samples had considerably higher levels of EAA except for histidine and tryptophan (Table 2) with little differences between algae species. Additionally, compared to the reference protein (egg), algae samples had relatively high levels of essential amino acids except for methionine and valine, indicative of fairly good protein quality (Table 2). This is reflected in the EAA indexes, which were higher for the algae samples (range 0.88 to 0.97) compared to wheat flour (0.66) and much closer to egg protein. The EAA indexes for TcR (0.89) and CvR (0.9) were similar to previously reported values [4], while the EAA index for MgR (0.88) was lower than the previously reported for *Nannocloropsis* sp. (1.02) [6]. Ethanol treatment increased the content (g/100 g protein) of most essential amino acids except for lysine (decrease for all three) and histidine (decrease in CvT). Consequently, ethanol-treated algae had slightly higher EEA indexes for all three algae but resulted in lower lysine levels (17–22% reduction). Lysine reduction was lowest for Mg (16.6%) and MgT had 5.26 g lysine per 100 g protein which was the highest lysine content of the ethanol-treated samples. Lysine is the most susceptible amino acid to process-induced chemical changes [34,35]. Ethanol treatment and subsequent drying, de-hydration and heat may have resulted in the observed lysine losses. Lysine was the limiting EAA in wheat flour with a DIAAS value of 46% (Table 2). In the untreated algae samples (TcR, CvR, MgR), valine (TcR) or methionine (CvR, MgR) were the limiting EAA, and DIAAS values ranged from 54 to 73% (Table 2), indicating that all three algae can potentially improve the protein quality of bread. After ethanol treatment lysine was the limiting amino acid in both TcT and CvT, however, DIAAS values were still considerably higher than for wheat flour with 78 and 73% respectively. MgT had the highest DIAAS of all algae samples with 84% and methionine, not lysine, as the limiting amino acid.

The three algae species had similar *in vitro* protein digestibility (Table 3). De-hydration by ethanol treatment and subsequent drying significantly reduced protein solubility during simulated digestion. Solubilized protein was extensively hydrolysed during digestion. After 120 min of simulated small intestinal digestion, approximately 90% of the soluble protein was in the form of small peptides (<1 kDa). Ethanol treatment affected the degradation of the solubilized proteins only marginally, with a very slight reduction of small peptides (87–82%). However, the difference in solubility induced by ethanol treatment resulted in a reduced protein digestibility (D_SEC_) of the treated vs the raw samples. This points towards an incomplete re-hydration of the dried treated samples during *in vitro* digestion. Among the raw algae samples, TcR had a lower protein digestibility compared to CvR and MgR, which was still evident after ethanol treatment. The D_SEC_ values reported here are generally lower than *in vitro* protein digestibility values obtained by the multi-enzyme method [4] which ranged from ~82–87% in the mentioned study. Such differences are not surprising given the huge differences in both simulated digestion and analytical quantification methods and such differences due to method have been previously reported [24,36]. Relative differences between the samples suggest that Cv and Mg have a higher protein digestibility than Tc and ethanol treatment had a negative influence on protein digestibility due to a reduction of protein solubility. Among the ethanol-treated samples, MgT had the highest *in vitro* protein digestibility.

### 3.3. Rheological Properties of Microalgae Substituted Wheat Doughs

Farinograph WA and DST for the wheat flour (control) and algae substituted (12% wheat flour substitution level) doughs is shown in Figure 1a,b. Wheat flour substitution with algae increased WA. This increase was highest for Cv, both in its raw (CvR), and ethanol-treated form (CvT) but was not significant for TcR and MgR. Tc and Mg both showed an increase in WA with ethanol treatment (significant for Mg). The high protein and fibre content of the algae, especially those treated with ethanol, can explain the observed increase in WA as both protein and fibres have been shown to increase WA of wheat flour doughs [8,37,38,39]. However, Farinograph WA is essentially a measurement of dough consistency during kneading and other components of the microalgae, such as fat and minerals may also have contributed to a difference in WA. DST significantly decreased with wheat flour substitution except for CvT. Doughs prepared with 12% CvR had an extremely low DST, which was considerably improved by ethanol treatment. However, ethanol treatment had no effect on DST for Tc and Mg. The observed decrease in DST upon wheat flour substitution can be partly explained by a dilution of gluten proteins, but other aspects, such as an interference of algae components with gluten network development also play a role as the different algae samples influenced DST differently. These differences may be linked to compositional differences, however, the number of samples (six different) in this study is too few to deduct a general relationship.

Besides a decrease in DST, the extensograph maximum resistance to extension (Figure 1c) and extensibility (Figure 1d) reveal a weakening of the dough structure due to wheat flour replacement with algae. All doughs with algae had significantly lower extensibility, while Rmax was not significantly different from the control dough for CvT and MgR. Ethanol treatment improved extensibility for all three strains (significant for Cv and Mg) and showed a tendency of increased Rmax for Tc and Cv, while the opposite occurred for Mg. Among the raw algae samples, MgR showed the best rheological properties (DST, Rmax, extensibility), while TcT and CvT generally performed better than MgT.

Also, the creep recovery test revealed a weakening of the dough structure with wheat flour substitution as both maximum compliance (Jmax) and elastic compliance (Je) decreased for algae substituted doughs (Figure 1e). Among the raw samples, TcR performed best with a Je similar to the control, while CvR and MgR both had significantly lower Je and Jmax compared to the control. Ethanol treatment increased Jmax and Je for all algae samples, but only TcT and MgT showed a significant improvement and a Jmax comparable to the control.

Substitution of wheat flour with 12% algae weakened the dough structure, which was evident in decreased DST, extensibility, resistance to extension, as well as decreased compliance (maximum compliance and elastic recovery compliance) in creep recovery tests. This can be explained by the dilution of gluten proteins, which is frequently pointed out as a reason for dough strength reduction when sifted wheat flour is substituted by other ingredients [38,40]. However, even at the same substitution level (and, therefore, the same gluten dilution), the different algae samples impacted dough structure differently. Differences also depended on the type of rheological measurement. While CvT had a DST and Rmax similar to the control, the extensibility of CvT did not differ from extensibility of the other ethanol treated algae doughs; Jmax and Je of CvT were lower than for MgT and TcT. Ethanol treatment improved almost all measured dough rheology parameters for Cv and Tc as it led to an increase in DST (for Cv), an increase in Rmax (not significant) and extensibility (significant for both), as well as an increase in Jmax and Je (significant for Tc). For Mg, on the other hand, ethanol treatment decreased DST and Rmax, but increased extensibility, Je and Jmax.

The molecular structure of the gluten network is quite complex and both covalent (disulphide bonds) and non-covalent (hydrogen bonds, hydrophobic and electrostatic interactions and chain entanglement) intermolecular interactions are important factors influencing the viscoelastic behaviour of gluten [41,42,43,44]. These can all be influenced by different constituents in the algae biomass and more detailed chemical analysis of a much higher number of different algae strains alongside their performance in wheat flour dough is needed to start unravelling the impact of each constituent on the gluten network and dough rheology.

### 3.4. Bread Quality

All bread containing TcR and CvR (12% substitution level) had intense dark green crusts almost verging on black with a high sheen, while bread with MgR also had dark green crusts, but less glossy and browner (Figure 2). Removal of chlorophylls by ethanol treatment resulted in lighter coloured bread crusts (Figure 2). Bread with TcT still had a distinctly green crust, which was much lighter and considerably less glossy than for corresponding TcR breads. Bread with CvT had the darkest crust amongst all bread prepared with ethanol-treated algae, but with a more brown and less green colour compared to TcT. Incorporation of ethanol-treated Mg resulted in bread with a pleasant brown crust reminiscent of a typical wholemeal bread. The bread crumb colour (Table 4) corresponded closely to the crust colour. All bread prepared with ethanol-treated algae had significantly higher L* values vs the bread prepared with untreated algae (Table 4). Ethanol treatment reduced the green hue to a greater extent from the crumbs of bread prepared with CvT and MgT (shown by a*-score), while some green colour remained for TcT (negative a* score). The degree of yellow (b*) was significantly different only between bread prepared with TcR and MgT (Table 4). Already the untreated form MgR showed a higher proportion of yellow compared to TcR and CvR, which was further increased during ethanol treatment (Table 4). The ∆*E* (overall colour difference) compared to the control demonstrated values >5 indicating the colour difference was perceivable by the human eye [45]. Overall, ∆*E* values were higher for the bread prepared with raw (TcR, CvR and MgR) compared to the corresponding bread prepared with ethanol treated (TcT, CvT and MgT) algae with MgT showing the lowest score, i.e., its colour differed the least from the control

Substitution of wheat flour with 12% algae biomass significantly decreased specific volume and significantly increased crumb firmness (Figure 3). Bread prepared with TcR or CvR was very firm and differed significantly from all other bread. Ethanol treatment significantly improved crumb firmness and specific volume for Tc and Cv, but not Mg. The bread prepared with MgR was different from the other bread with raw algae biomass as they had a much softer crumb and higher specific volume. Among the bread prepared with ethanol-treated biomass TcT had the lowest crumb firmness and highest specific volume. The negative effects of wheat flour substitution on bread characteristics can be explained by a weakening of the dough structure as demonstrated by dough rheology. Different rheology parameters estimating dough strength showed strong (r > 0.75) to weak (r < 0.50) positive correlation coefficients with bread volume. Correlation coefficients of 0.76 for Rmax, 0.87 for extensibility, 0.7 for DST and 0.49 for Je were found. Correspondingly a strong to moderate negative correlation coefficients with crumb firmness were noticed for extensibility, DST and Je with coefficients of, −0.78, −0.65 and −0.65, respectively. However, between Rmax and crumb firmness no correlation was found (r = −0.36). Extensibility showed the strongest correlations with baking performance (bread volume and firmness). This agrees with previous studies suggesting that large-scale deformation tests, such as extensograph are well suited to predict the baking performance of different wheat flours. A positive correlation between bread volume and Je for different wheat flours has also been reported [46,47]. However, in this study baking performance of algae substituted wheat flour doughs was better described by extensograph than creep recovery tests.

All bread prepared with raw algae received negative scores for the three sensory attributes (smell, appearance, and colour) (Figure 4) in the preliminary sensory evaluation. The panel disliked the smell attributes the most which received the most negative score. Ethanol treatment improved the acceptability of the bread with positive scores for appearance and colour. The rating of colour by the panel is consistent with the ones measured instrumentally, reported as ∆*E* (Table 4). Breads with ethanol-treated Tc received a negative score for smell, CvT was rated neutral, while MgT received a positive score. MgT was also the sample with the highest score in colour. One of the main hurdles in the incorporation of algae into bread is the negative impact on the sensory properties of the bread and especially the green colour [9]. Incorporation levels of 3% algae are usually enough to bring noticeable changes in sensory properties of bread [9,48]. It is, therefore, not surprising that the bread prepared with 12% untreated algae acquired very low scores for sensory attributes. Ethanol extraction clearly increased the likeability of the algae substituted bread. Besides the removal of chlorophyll, which improved the colour scores, ethanol treatment reduces volatile compounds, which predominantly cause the unpleasant smells [13,49]. Volatile compounds in untreated microalgae biomass are dominated by sulphur compounds, [9], aldehydes and ketones [13]. These compounds are usually removed up to ~80% by the ethanol treatment, as shown in our previous study [13].

### 3.5. Amino Acid Composition and Assessment of Protein Quality in the Microalgae

Substitution of wheat flour by 12% microalgae increased the protein content of the bread between 11–36% compared to the control (Table 5). Since wheat flour substitution increased Farinograph water absorption, bread containing microalgae were prepared with more water compared to the control bread and retained a higher moisture content after baking. Due to this higher moisture content, the protein content of the bread as eaten (per 50 g bread portion) was somewhat lower than calculated based on the dry weight. Due to the higher protein content of the ethanol-treated algae, bread prepared with TcT, CvT and MgT showed a higher increase in protein content (25–36%) compared to the corresponding TcR, CvR and MgR (11–14%). The increase in protein content was >30% in CvT and MgT which makes them qualify for the “increased in protein” nutrition claim set by the EFSA.

Substitution of wheat flour with algae also had a positive effect on protein quality. Therefore, against the control, the EAAI score for all the bread with microalgae addition was higher with MgT based bread depicting the greatest score consistent with the results of MgT powder reported in Table 2.

Lysine was the first limiting amino acid in all mixtures. The DIAAS score in the control was the lowest (46%). Due to the higher content of EAA, including lysine, in microalgae DIAAS values increased to 57% (CvT), 60% (TcT), 63% (MgT), 65% (TcR), 63% (CvR) & 66% (MgR). Processing can influence the bioavailability of lysine (e.g., through thermal degradation), but the amino acid composition of the bread was similar to that of the ingredient mix and no degradation of lysine during baking was observed (Table 5).

Different protein sources and processing techniques also impact protein digestibility, which is an important part of protein quality. Ideally, protein digestibility should be determined in vivo as true ileal digestibility of individual amino acids. However, this is time-consuming, invasive, or requires the use of dual-isotope labelled proteins, which makes it impossible to apply for screening purposes or to understand the effect of food processing on protein digestibility. There is, therefore, a range of different *in vitro* methods to estimate protein digestibility. The method applied in the current study is based on the INFOGEST static *in vitro* digestion model and estimated protein digestibility by quantifying the proportion of protein that is solubilised (using Dumas) and degraded into small peptides (using SEC) during simulated digestion. As seen for the microalgae powders (Table 2), the bread prepared with microalgae showed a large enzymatic degradation of proteins into small peptides (82–87% of dissolved protein), which was equal to or higher than for the control (82.7%) (Table 6). However, bread prepared with microalgae, especially microalgae treated with ethanol, showed decreased protein solubilization during digestion (59–74%), while protein solubilization from the wheat control was high (83.2%) (Table 6). This resulted in a decreased *in vitro* protein digestibility (D_SEC_) in % of total protein for the algae-containing bread compared to the control bread (Table 6). This was more pronounced for the bread containing ethanol-treated algae. However, these breads also contained higher protein levels. Clearly, more research is needed to understand and improve protein digestibility of microalgae enriched bread. The applied methodology for measuring protein solubilization based on measurement of soluble nitrogen might not have been optimal for the microalgae containing bread and dedicated analytical tools for microalgae proteins should be developed to increase the understanding of their contribution to human nutrition in complex food systems.

## 4. Conclusions

Three algae species differed clearly in their effect on dough rheology and baking performance. For these attributes, MgR performed better than CvR and TcR but still scored low on the sensory properties of bread. Thus, consumer acceptable bread with MgR might be achieved at lower addition levels. However, the high crumb firmness and low bread volume of TcR and CvR bread, together with the unpleasant smell of these two algae, hinder their application in wheat bread for protein enrichment. Ethanol pre-treatment of microalgae improved dough rheology (Rmax, extensibility and Je), baking performance and sensory properties for Tc and Cv, but for Mg only the smell and colour of the bread were improved. Replacement of wheat flour with 12% microalgae increased protein content and improved the protein quality of the bread. While ethanol treatment increased the protein content of the microalgae, it resulted in relatively lower levels of lysine and lower *in vitro* protein digestibility. The lower protein digestibility was due to a reduced protein solubility and was still evident in the fortified bread. However, the improved amino acid composition and increased protein content of the fortified bread compensated for the reduced protein digestibility. Our results suggest that ethanol treatment, or other processing methods of microalgae that can remove compounds that lead to undesirable smell, colour, or taste in final breads is necessary to achieve nutritionally relevant fortification. More research should be directed into the selection of appropriate microalgae species for specific food applications alongside the development of tailored pre-processing methods. We also noticed that *Microchloropsis gaditana* combined with ethanol treatment to be a good candidate for protein fortification of wheat bread.

## Figures and Tables

**Figure 1 foods-10-03078-f001:**
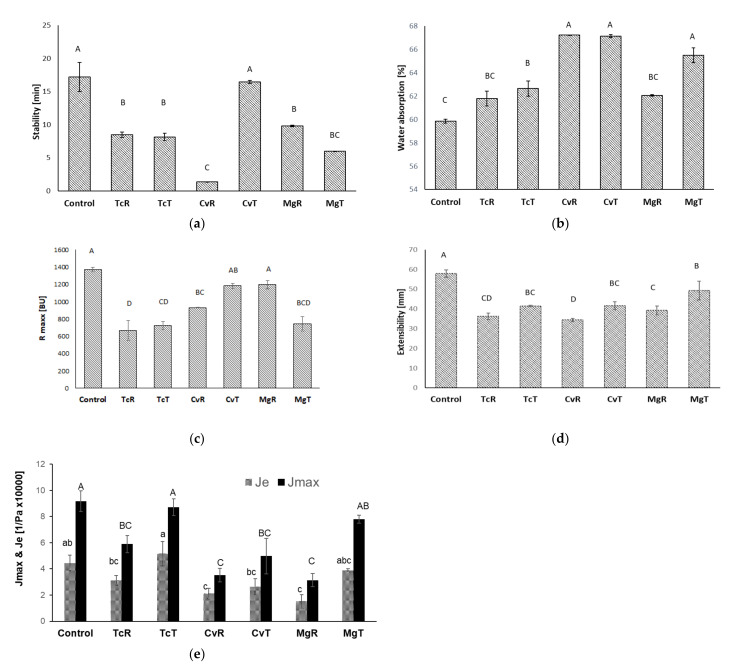
Farinograph dough stability time (**a**), water absorption (**b**); Extensograph maximum resistance to extension (**c**), extensibility (**d**) and rheometer maximum creep compliance (Jmax) and elastic recovery compliance (Je) (**e**). Data are mean values of duplicate measurements +/− standard deviation. Bars sharing the same letter are not significantly different (*p* > 0.05).

**Figure 2 foods-10-03078-f002:**
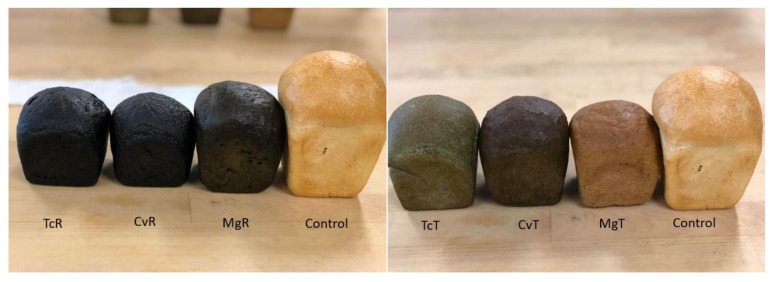
The appearance of bread prepared with 12% TcR, TcT, CvR, CvT, MgR or MgT and the control bread (100% wheat).

**Figure 3 foods-10-03078-f003:**
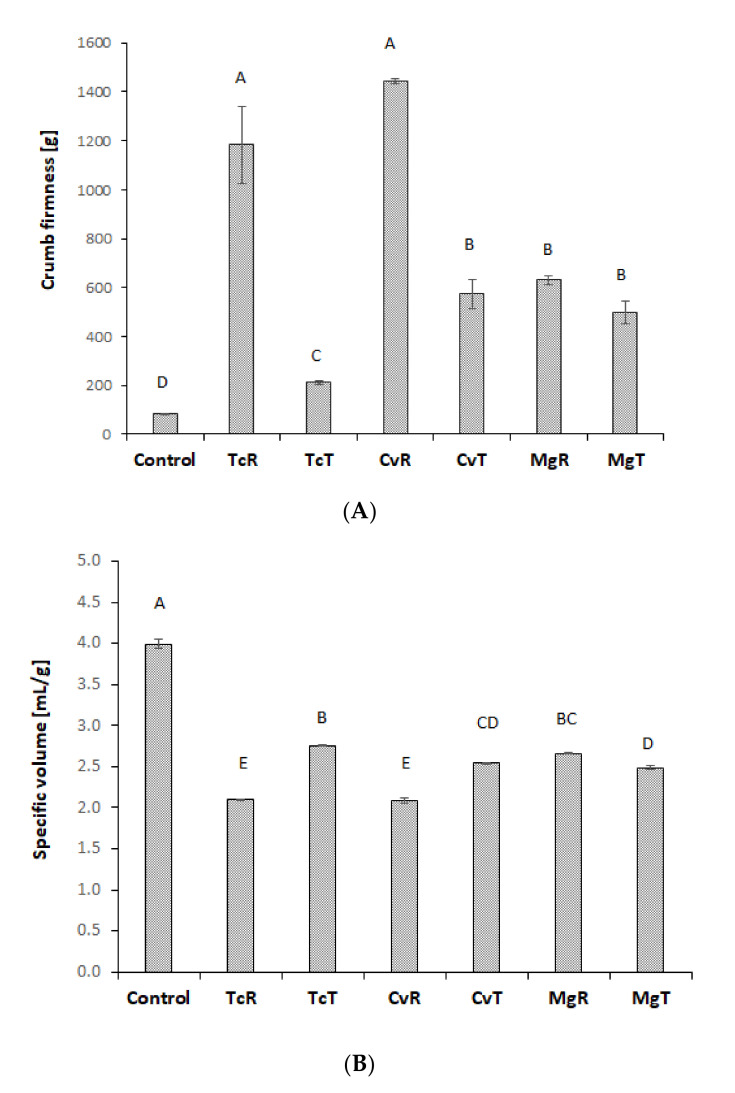
Crumb firmness (**A**) and specific volume (**B**) of bread prepared with 12% TcR, TcT, CvR, CvT, MgR or MgT and the control bread (100% wheat). Data are mean values of two independent baking trials with three small breads each (*n* = 6). Bars sharing the same letter are not significantly different (*p* > 0.05).

**Figure 4 foods-10-03078-f004:**
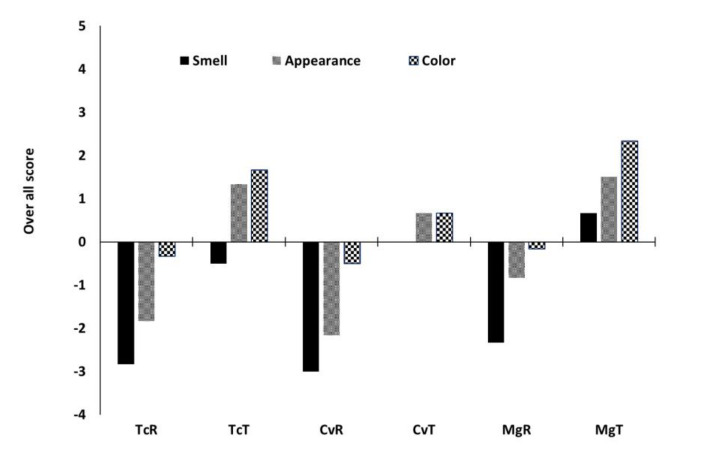
Preliminary estimates of sensory properties of breads prepared with 12% microalgae. Data are the average score of 6 non-trained panellists ranking the bead smell, appearance, and colour between −4 (disliked extremely) and 4 (like extremely).

**Table 1 foods-10-03078-t001:** Macronutrient composition in g/100 g dry weight and colour of raw (R in abbreviation) and ethanol treated (T in abbreviation) microalgae (*Tetraselmis chui, Chlorella vulgaris* and *Microchloropsis gaditana*) samples and wheat flour.

	Wheat Flour	TcR	TcT	CvR	CvT	MgR	MgT
Protein	13.5 ± 0.3	42.1 ± 0.1	59.5 ± 0.2	47.8 ± 1.1	58.8 ± 0.3	43.3 ± 1.5	61.7 ± 2.8
Free amino acids	n.a.	5.4	2.9	3.3	0.8	3.7	0.4
Fat	2.3 ^+^	13.8	0.3	15.7	0.6	21.6	0.4
Dietary fibre	3.8 ^+^	8.9 ± 0.8	15.1 ± 2.2	13.8 ± 0.5	19.0 ± 0.6	12.2 ± 0.6	21.8 ± 0.8
Ash	0.6–0.75 ^+^	16.0 ± 0.1	16.7 ± 0.1	6.7 ± 0.6	8.2 ± 0.0	7.0 ± 1.1	7.4 ± 0.1
L*	92.3 ± 0.1	20.9 ± 1.3	51.6 ± 3.4	12.9 ± 1.0	48.3 ± 3.6	22.7 ± 0.6	58.0 ± 0.9
a*	0.48 ± 0.1	−10.1 ± 1.0	−5.6 ± 0.2	−1.3 ± 0.4	6.6 ± 0.5	−4.4 ± 0.1	3.6 ± 0.3
b*	10.8 ± 0.3	15.4 ± 1.9	17.6 ± 0.7	17.0 ± 0.6	23.6 ± 0.9	10.2 ± 0.4	29.3 ± 0.4
∆*E* (compared to wheat flour)		72.2	41.6	79.6	46.4	70.1	39.2

+ Values taken from product data sheet. n.a. = not available.

**Table 2 foods-10-03078-t002:** Amino acid profiles (g/100 g protein) and protein quality (EAAI, DIAAS) of microalgae (*Tetraselmis chui, Chlorella vulgaris* and *Microchloropsis gaditana*) compared to wheat flour and egg protein (reference).

Amino Acid ^1^ (g/100 g Protein)	Wheat Flour	TcR	TcT	CvR	CvT	MgR	MgT	Egg ^2^
Alanine	2.8	8.6	7.1	9.5	8.1	6.7	7.2	-
Arginine	3.5	4.5	5.1	6.3	6.9	5.7	6.5	6.2
Aspartic acid	3.9	11.9	12.5	8.9	9.6	8.6	9.6	11.0
Cysteine	2.6	2.0	1.2	1.1	0.8	1.0	1.0	2.3
Glutamic acid	33.8	12.8	13.5	10.5	11.5	10.7	12.0	12.6
Glycine	3.5	6.4	6.6	6.5	6.2	5.9	6.3	4.2
Histidine	2.0	1.7	1.7	1.8	1.7	1.9	2.0	2.4
Isoleucine	3.1	4.0	4.4	4.0	4.4	4.1	4.9	6.6
Leucine	6.7	8.5	8.9	9.4	10.2	8.6	9.9	8.8
Lysine	2.4	6.2	4.9	5.7	4.5	6.3	5.3	5.3
Methionine	1.9	2.4	2.5	1.9	2.2	2.0	2.3	3.2
Phenylalanine	5.1	5.4	5.9	5.3	5.7	4.6	5.3	5.8
Proline	12.2	5.4	4.4	8.9	5.2	14.2	5.7	4.2
Serine	5.1	4.6	4.8	4.7	4.7	4.2	4.8	6.9
Threonine	2.7	4.6	4.9	4.7	5.0	4.7	5.2	4.0
Tryptophan	1.9	1.8	2.1	1.9	2.3	1.9	2.3	1.7
Tyrosine	2.9	3.4	3.3	3.4	3.6	3.3	3.6	4.2
Valine	3.8	5.9	6.1	6.0	6.4	5.3	6.0	7.2
EAAI	0.74	0.89	0.91	0.90	0.93	0.88	0.97	1.00
DIAAS (%) ^3^	45 (Lys)	54 (Val)	78 (Lys)	69 (Met)	73 (Lys)	73 (Met)	84 (Met)	

^1^ Difference between parallels <6% for each individual amino acid. ^2^ Reference values were taken from Becker (2007) [1]. ^3^ the limiting amino acid is indicated in parentheses.

**Table 3 foods-10-03078-t003:** *In vitro* protein digestibility of microalgae (*Tetraselmis chui*, *Chlorella vulgaris* and *Microchloropsis gaditana*) ingredients.

*In Vitro* Protein Digestibility (%)	TcR	TcT	CvR	CvT	MgR	MgT
Dissolved protein	57.5 ± 2.7 ^b^	33.7 ± 1.5 ^c^	68.5 ± 1.3 ^a^	52.9 ± 5.2 ^b^	68.5 ± 3.7 ^a^	53.6 ± 0.2 ^b^
Small peptides	90.3 ± 0.4 ^a^	87.2 ± 1.3 ^b^	89.4 ± 0.2 ^a^	82.3 ± 0.3 ^c^	90.8 ± 0.2 ^a^	87.4 ± 0.4 ^b^
*In vitro* protein digestibility as D_SEC_	51.9 ± 2.7 ^b^	29.4 ± 1.5 ^d^	61.2 ± 1.0 ^a^	43.5 ± 4.2 ^c^	62.1 ± 3.5 ^a^	46.8 ± 0.3 ^bc^

Values in the same row sharing the same letter are not significantly different (*p* > 0.05).

**Table 4 foods-10-03078-t004:** Bread crumb colour (mean, *n* = 3 Mean ± SD) with microalgae (*Tetraselmis chui, Chlorella vulgaris* and *Microchloropsis gaditana*) biomass.

	Control	TcR	TcT	CvR	CvT	MgR	MgT
L*	74.1 ± 3.1 ^a^	24.3 ± 1.0 ^f^	45.0 ± 0.5 ^c^	25.3 ± 0.5 ^f^	38.3 ± 1.4 ^d^	33.0 ± 0.5 ^e^	55.6 ± 1.2 ^b^
a*	0.4 ± 0.3 ^d^	−1.0 ± 0.5 ^e^	−2.7 ± 0.0 ^f^	2.0 ± 0.1 ^c^	2.5 ± 0.9 ^b^	1.9 ± 0.3 ^c^	5.7 ± 0.2 ^a^
b*	14.9 ± 0.7 ^a^	−2.5 ± 0.4 ^e^	15.4 ± 1.2 ^c^	−1.5 ± 0.3 ^e^	11.4 ± 1.5 ^d^	8.4 ± 0.6 ^d^	22.9 ± 0.3 ^b^
∆*E*		52.6	29.1	51.4	35.9	42.5	20.7

∆*E* calculated from average values of the three replicates. Numbers in the same row sharing the same letter are not significantly different (*p* > 0.05).

**Table 5 foods-10-03078-t005:** Amino acid profiles (g/100 g protein), protein content per portion of bread (50 g), moisture content of bread, protein quality (EAAI, DIAAS) of microalgae (*Tetraselmis chui, Chlorella vulgaris* and *Microchloropsis gaditana*) bread compared to control wheat flour bread and egg protein.

Amino Acid ^1^ (g/100 g Protein)	Control	TcR	TcT	CvR	CvT	MgR	MgT	Egg ^2^
Alanine	3.4	5.1	4.9	4.9	4.8	4.0	4.5	-
Arginine	3.2	3.3	3.6	4.2	4.7	4.2	4.6	6.2
Aspartic acid	3.6	5.9	6.7	5.6	6.4	5.2	6.1	11
Cysteine	2.3	2.0	2.0	1.9	1.8	1.9	1.7	2.3
Glutamic acid	34.2	28.1	26.8	27.8	27.1	29.1	27.1	12.6
Glycine	3.8	4.6	5.0	4.4	4.6	4.2	4.6	4.2
Histidine	2.1	2.1	2.0	2.0	1.9	1.9	2.0	2.4
Isoleucine	3.7	4.0	4.2	3.7	3.8	3.7	4.0	6.6
Leucine	6.7	7.5	7.8	7.5	8.0	7.3	8.2	8.8
Lysine	2.4	3.4	3.5	3.3	3.0	3.3	3.6	5.3
Methionine	1.4	1.7	1.7	1.9	2.0	1.7	2.0	3.2
Phenylalanine	5.3	5.5	5.6	5.5	5.3	5.2	5.6	5.8
Proline	12.6	10.0	9.5	10.5	9.1	12.1	8.7	4.2
Serine	4.7	4.9	4.9	4.5	4.5	4.3	4.5	6.9
Threonine	2.4	3.3	3.4	3.1	3.4	3.3	3.6	4.0
Tryptophan	1.7	1.4	1.6	1.6	1.7	1.6	1.6	1.7
Tyrosine	2.1	2.2	1.8	2.8	3.0	2.4	2.7	4.2
Valine	4.3	4.9	5.1	4.9	5.1	4.6	4.9	7.2
Protein (N × 5.7) g/100 g freeze-dried bread	13.14 ± 0.05	16.69 ± 0.05	18.31 ± 0.13	17.78 ± 0.03	19.16 ± 0.25	16.69 ± 0.1	19.06 ± 0.03	
Protein (g/50 g bread)	4.5 ± 0.02	5.0 ± 0.01	5.7 ± 0.04	5.1 ± 0.01	6.0 ± 0.08	5.2 ± 0.03	6.2 ± 0.01	
% increase in protein rel. to control		10.5	25.3	13.2	32.1	14.4	35.6	
Moisture content (%)	36.7 ± 0.4	41.2 ± 2.1	40.0 ± 1.4	43.2 ± 1.1	39.8 ± 1.5	41.9 ± 0.2	40.7 ± 1.3	
EAAI	0.67	0.73	0.73	0.72	0.74	0.71	0.77	1.0
DIAAS (%) ^3^	46	65	60	63	57	66	63	

^1^ Relative standard deviation between parallels <6% for each individual amino acid. ^2^ Reference values were taken from Becker (2007) [1]. ^3^ Lysine was the first limiting amino acid among all the formulations.

**Table 6 foods-10-03078-t006:** *In vitro* protein digestibility of breads prepared with 100% wheat flour (control) or wheat flour substituted with 12% microalgae (*Tetraselmis chui, Chlorella vulgaris* and *Microchloropsis gaditana*) biomass.

*In Vitro* Protein Digestibility (%)	Control	TcR	TcT	CvR	CvT	MgR	MgT
Dissolved protein	83.2 ± 1.4 ^a^	67.4 ± 1.8 ^c^	59.3 ± 1.4 ^d^	74.2 ± 1.2 ^b^	67.6 ± 1.7 ^c^	70.1 ± 1.6 ^bc^	61.7 ± 0.4 ^d^
Small peptides	82.7 ± 0.8 ^d^	86.9 ± 0.1 ^ab^	86.2 ± 0.1 ^a^	85.5 ± 0.1 ^bc^	84.8 ± 0.5 ^c^	86.0 ± 0.3 ^ab^	86.2 ± 0.3 ^ab^
*In vitro* protein digestibility as D_SEC_	68.5 ± 0.6 ^a^	56.1 ± 2.8 ^c^	50.5 ± 0.6 ^d^	63.4 ± 1 ^b^	57.3 ± 1.2 ^c^	60.3 ± 1.3 ^bc^	53.2 ± 0.7 ^d^

Values in the same row sharing the same letter are not significantly different (*p* > 0.05).

## Data Availability

The data presented in this study are available on request from the corresponding author.

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
