# Peer review of "Protein Enrichment of Wheat Bread with Microalgae: Microchloropsis gaditana, Tetraselmis chui and Chlorella vulgaris"

_foods, 2021, doi:10.3390/foods10123078_

Round 1
Reviewer 1 Report
In my opinion it is a very interesting study which fits well to the scope of Foods Journal. Additionally microalgae is of an interest of researchers and food industry so I think the manuscript will reach many readers.
Several comments and questions included below:
What did you use for icing the breads?
In some parts of the manuscript I have noticed double spaces between the words, please correct it.
Line 14: CV without slash
Line 43: “microalga”? check the spelling
Line 49: EFSA it is obvious for people from Europe what does it mean but in case write a full name of the institution in brackets like in the line 68 but introduce a full name when you are mentioning it for the first time
Line 218 and 283: remove a dot (“.”) at the end of heading
Line 200: The abbreviation “EAAI” is clear to me but may not be to everyone so please provide a full name in the brackets.
Table 1 you are using different font style.
In my opinion the Table 1, 2, 3 should be improved by providing a standard deviation and homogenic groups. Additionally it should be easy to read without the need to find the meaning of the abbreviations in the text, so provide it at least in the heading of the table.
Line 315: what does the question mark means?
Line 342, 427: vs with a slash
I suggest to improve the graphical quality of Figure 4, also showing a standard deviation on the bars.
Line 552: I think the statement “dramatically improved” is a bit too strong at the light of the results.
Line 555: What do you mean by ” improved protein quality” ?
All in all, I think the manuscript is very good and can be accept after minor changes.
Reviewer 2 Report
The investigation to produce novel foods incorporating alternative natural protein sources such as microalgae is a topic of great interest and of high importance as the increasing population leads to depletion in resources. The authors investigated the incorporation of three different microalgae with or without ethanol pretreatment in wheat bread production and studied their effect on several quality parameters.
In general, the reading of the manuscript is linear, and the length is appropriate. It is seemed that a lot of work has been done, nevertheless some major issues have to be considered prior publication.
Major issues
The results are interesting; however, I would like the authors to refer, in the statistical analysis section, the replicates of the experiment (vats), as well as the replicates for the experimental treatment. If there is only one replicate breadmaking vat, I am concerned about the validity of the trends that the authors have identified and discussed. In breadmaking experiments, the experimental unit is the vat. Repeating assays does not add accuracy to the experiment as most variance come from breadmaking, not from analysis.
Please refer in the text the novelty of the investigation instead of other studies already existed in the literature.
Minor issues
Please carefully check the whole manuscript for errors. Correct the following:
Use subscripts and superscripts wherever is needed (example lines 100, 106, 119, 138…).
Add a space between number and unit except for degrees Celsius.
Change ml to mL.
Lines 164-166. What do you mean ‘’was diluted to ca. 21% dm/100 g dm’’? Please explain or correct.
Lines 168-170. ‘’ Cell wall disruption efficiency in Mg is found…’’ Is this a statement found in the literature? If yes please add at the end of the sentence the reference, if it is an experimental observation change the word ‘’is’’ to ‘’was’’.
Line 179. Is there a specific reason, you use two different drying techniques? The use of high temperatures for spray drying may lead to extracts of lower quality instead of freeze-drying technique.
Lines 186-187.The word respectively does not correspond somewhere. Please correct.
Line 196. The sentence is not appropriate to begin with a number. Please add words (example. An amount of..).
Line 220. Correct ‘’were evaluated’’ to ‘’was evaluated’’.
Line 242. Add full stop at the end of the paragraph.
Lines 251; 258; 264. You refer ‘’as previously described’’. Please add after the word ‘’described’’ in brackets the paragraph of the manuscript that the techniques have been described, otherwise add a short description.
Lines 263-267. Please describe the color parameters L, a and b and the equation.
Line 284. Table 1. Please provide standard deviation values to evaluate the significances. Please check the value of parameter a of the TcR, you may correct it to (-10.1).
Lines 308-309. In Table 2 it is seemed that except histidine, also proline, serine and tryptophan of the studied algae are presented in lower levels compared to wheat flour. Please explain.
Lines 316-317. In Table 2 it is seemed that except lysine and histidine, also proline is not higher in ethanol treated samples compared to the non-treated. Please explain.
Line 365. Remove word ‘’time’’, after DST.
Figure 1. Please check the error bars and the ANOVA as some error bars are not in accordance with the presented letters.
Figure 1d. Replace small letters with uppercase letters.
Lines 374-377. Please check carefully and correct the legend as the given numbers does not correspond to the figures.
Lines 445-447. Please give the legend in a completed form. Crumb firmness and specific volume of….
Lines 482-486. I think that the incorporation level of 3% is referred to the total. Your incorporation (12%) is referred to the wheat flour. How much was your incorporation to the total? Please clarify.
Lines 493-503. I recommend the results for the protein content to be expressed in dry matter in order the data to be comparable in a more scientific base. Please add in Table 5 the protein content in dry matter and calculate the %increase in protein compared to control.
Reviewer 3 Report
- GENERAL:
- The degree of the scientific novelty of the article is acceptable. The article value compared to that of other articles on the topic is acceptable;
- English style must be reviewed in depth in whole text. Check for these through the text. A native English speaking person should review the revised manuscript before a new submission;
- please remove all double spaces throughout the text;
- All units should be checked (example: page 3, line 105 – change to 25-27 °C; page 3, line 108 - 3000g change to 3000 g; page 3, line 109 – 15-20% change to 15-20 % t.c. - check in whole text);
- TITLE: ok
- ABSTRACT:
- You should standardize italicized names throughout the text;
- KEYWORDS: Modify second and third keywords;
- INTRODUCTION: ok;
- MATERIALS AND METHODS:
- Page 4 line 165 - % or g/100g???;
- Page 5, line 216 - Change „All analyses are mean of at least duplicate measurements” for example to “At least duplicate measurements were performed for all analyzes”;
- Page 5, line 238 - expand the shortcut (SEC);
- Page 6 line 263 – Should be: bread crumb;
- Page 6, line 269 – In my opinion there was to few panel participants; Additionally Sensory assesment should be changed for example to Bread quality evaluation; Whether the samples were coded?
- Page 6, line 276 - How many samples per treatment, per replication? In my opinion 2 batches could be not enough to proper analysis;
- Page 6, line 279 - Please note the use of correlation analysis is fraught with danger - correlation only tells you the association between traits, not that one trait influences the other or is causative of changes in the other. So, they need to be carefully used and in addition if you do an experiment and create different treatments then pool the data and look at the correlation you can get a false idea of the strength of the association because of the variance due to treatments.
RESULTS AND DISCUSSION:
- Figure 1 - uppercase and lowercase letters are used - please standardize it;
- statistical analysis data should be carefully checked (tables and graphs do not show statistically significant differences - no letters in the superscript - Table 4; no deviations - Figure 3; incorrectly assigned letters in the bars - Figure 3, etc.)
- There are some poorly constructed sentences and incorrect subject/verb agreement that make meaning of sentences unclear;
- Page 15, line 553 – delete „dramatically“
- - Page 10, line 391 - rewrite the sentence (..improvement was low and not significant … - so there was no improvement?);
- Page 10, line 404 – the same;
- FIGURES: Please check all Figures;
- TABLES: Please check all Tables;
Reviewer suggest authors better rewrite the paper comprehensively before publishing it.
Round 2
Reviewer 2 Report
I think that the authors have addressed satisfactorily the comments of the reviewers. I consider that this manuscript may be published in its present form.
Reviewer 3 Report
Accept in present form.